# Confocal Microscopy for Intraoperative Margin Assessment of Lumpectomies by Surgeons in Breast Cancer: Training, Implementation in Routine Practice, and Two-Year Retrospective Analysis

**DOI:** 10.3390/cancers17172852

**Published:** 2025-08-30

**Authors:** Irene Cattacin, Timothée Rochat, Anis Feki, Arrigo Fruscalzo, Michel Boulvain, Benedetta Guani

**Affiliations:** 1University Hospital of Lausanne, CHUV, 1005 Lausanne, Switzerland; 2University of Fribourg, 1700 Fribourg, Switzerland; anis.feki@h-fr.ch (A.F.); arrigo.fruscalzo@h-fr.ch (A.F.);; 3University Hospital of Fribourg, HFR, 1752 Fribourg, Switzerland; timothee.rochat@h-fr.ch; 4University of Lausanne, 1015 Lausanne, Switzerland

**Keywords:** breast-conserving surgery, oncological surgery, confocal microscopy, margin assessment, Histolog Scanner

## Abstract

Breast-conserving surgery is a standard approach in early-stage breast cancer, aiming to remove the tumor while preserving healthy breast tissue. A key challenge is ensuring that the margins of the removed tissue are free of cancer cells, as involved margins often lead to repeat surgery. This study evaluates the use of the Histolog Scanner, a confocal microscopy device that provides real-time imaging of the surgical specimen during an operation. One surgeon underwent a structured training program and then used the scanner routinely over a two-year period. The findings demonstrate that the device enabled an accurate intraoperative assessment of the margin status and eliminated the risk of re-excisions in this cohort. This research suggests that with appropriate training, confocal microscopy can be effectively integrated into clinical practice to improve surgical precision and reduce the burden of repeat procedures for patients undergoing breast-conserving surgery.

## 1. Introduction

Each year in Switzerland, approximately 6,600 new cases of breast cancer are diagnosed. In total, 80% of affected women are 50 years or older at the time of diagnosis. Breast cancer remains the most prevalent cancer diagnosis among women in Switzerland, accounting for 32% of all female cancers [1]. The recommended treatment for early-stage disease is breast-conserving surgery (BCS), which aims to remove the tumor with a margin of normal tissue surrounding the resected specimen. BCS is always followed by radiotherapy as a complementary treatment in breast-conserving therapy, while chemotherapy is administered in certain cases, either before surgery (neoadjuvant) or after surgery (adjuvant), depending on the clinical context. Long-term outcomes of BCS are equivalent to mastectomy for early-stage breast cancers, provided that clear surgical margins are achieved [2,3] A recent meta-analysis suggests a survival advantage for women undergoing breast-conserving surgery with adjuvant radiotherapy for early-stage breast cancer compared with mastectomy [4,5]. Patients who underwent lumpectomy reported greater satisfaction with their breasts, psychosocial well-being, and sexual well-being [6,7,8,9].

Current ESMO consensus guidelines define a negative surgical margin as the absence of malignant cells at the surface of the resected invasive cancer specimen. The phrase “no tumor on ink” indicates that the inked borders are free from detectable tumor tissue. For ductal carcinoma in situ (DCIS), a clear margin of 2 mm is now recommended. When a positive margin (<2 mm) is detected, surgical re-excision becomes necessary [10,11]. According to the Quality Dashboard of Breast Cancer Centers in Switzerland, the proportion of BCS procedures should range between 70% and 90% for T1 tumors (≤20 mm), with at least 95% of resections achieving clear margins (R0), excluding cases of DCIS [12].

In the literature, approximately 20% to 40% of BCS procedures result in margins that are either positive or too close to the tumor, necessitating a re-excision, a second surgical procedure to remove additional tissue [13]. In our hospital, the re-excision rate prior to the study was 5%, consistent with the recommendations of the Swiss Cancer League accreditation criteria for Breast Centers in Switzerland. In many cases where re-excision is needed, patients opt for mastectomy in the second intervention to avoid the risk of a third surgery. Mastectomy, in addition to psychological and cosmetic damage, requires a longer hospital stay and increased postoperative care. The high re-excision rate leads to additional costs, increased patient anxiety, and a higher risk of postoperative complications. Additionally, the cosmetic result is degraded after re-excision. Therefore, improved intraoperative imaging tools are needed to ensure clear margins during BCS [13]. Several intraoperative margin assessment methods are available, but they have clinical and technical limitations preventing their widespread adoption. Intraoperative mammography of the lumpectomy specimen is commonly used, but its accuracy is limited by the inability to distinguish tumor tissue from dense fibro-glandular tissue [13]. Additionally, this method requires intervention from the radiology service, which is not available in all centers and can be time-consuming, especially if radiology facilities are distant from the operating room. Some centers use frozen sections to analyze margins, which requires the on-site presence of a pathologist. In our center, the surgeon examines an intraoperative mammogram of the lumpectomy specimen for non-palpable lesions to confirm the presence of the clip marker.

An innovative confocal laser microscopy scanner is now available for clinical use. The study by Elfgen et al. evaluated the Histolog^®^ Scanner (HS) for clinical application showing that it give access to the tissue morphology and cellular details of the superficial layers of fresh, thick tissue [14]. The authors concluded that the Histolog^®^ Scanner allows the detection of breast cancer in fresh human tissue with high accuracy and reliability. It was reported to be simple to use, cost-effective, and time-efficient, with great potential for routine intraoperative margin assessments. The HIBISCUSS study [15], conducted by Conversano et al., focused on the ability of the HS to provide real-time intraoperative imaging of breast tissue. The study showed that both surgeons and pathologists could quickly learn to differentiate between normal and cancerous breast tissue, with accuracy improving from 83% to 98%. The HELIXIR study [16], conducted by Togawa et al., investigated the use of the HS for intraoperative margin assessments compared to conventional radiography. The study found that the HS allows for the high-resolution, real-time evaluation of lumpectomy margins, which could potentially reduce the rate of re-excisions. The SENOSI study [17], conducted by Wernly et al., investigated the use of the HS for detecting positive margins in non-palpable breast cancer during surgery. Their findings emphasized that the HS enables high-resolution, real-time imaging, which improves the detection of cancerous cells at surgical margins and may help reduce the need for additional surgeries. In a recent article, the HS was used by pathologists for intraoperative assessments of lumpectomies. They showed that this approach identified all the positive margins, resulting in 100% sensitivity with an absence of re-excision in the included patients [18]. When used by breast surgeons, the detection rate was proposed to be linked to the level of understanding of HS images [19]. Last year, a new training program was developed based on the HS breast tissue image atlas [20], providing an advanced framework to train surgeons in intraoperative margin assessment using the HS. This training was completed by surgeons in the recent SHIELD study, resulting in an 81% detection rate of positive margins and a 67% reduction in the reoperation rate [21].

We hypothesized that, after appropriate training, surgeons could accurately perform intraoperative margin assessments of lumpectomy specimens using the Histolog^®^ Scanner. We further postulated that integrating this technique into routine practice would reduce re-excisions. After describing this new training program and the related learning curve of one surgeon, our article shows for the first time, the implementation of the HS in the routine practice of breast margin assessment. Clinical outcomes from a retrospective analysis of 68 consecutive patients corresponding to two years of practice are presented.

## 2. Materials and Methods

### 2.1. Study Design, Setting, and Population

This is a retrospective study with the primary aim of assessing the feasibility of introducing the Histolog^®^ Scanner (HS) for intraoperative use by surgeons during routine lumpectomies. Our focus was on evaluating the quality of the training process and the surgeons’ performance after training.

Data from consecutive adult female patients treated with breast-conserving surgery for breast cancer and intraoperatively assessed by one trained surgeon using the HS between December 2022 and January 2025 at the Department of Gynecology and Obstetrics of the University Hospital of Fribourg, Switzerland, were retrospectively analyzed. All included patients signed a general consent form allowing the reuse of their data for research purposes.

We excluded patients <18 years old, treated with non-conserving approaches or operated without the trained surgeon. Men were all treated with mastectomy and therefore excluded.

### 2.2. Confocal Microscopy and Implementation in Our Practice

We used the Histolog^®^ Scanner (HS), a medical device from SamanTree Medical SA, Switzerland, for the intraoperative margin assessment of breast lumpectomy specimens. The HS is a digital microscopy scanner for ultra-fast confocal imaging of freshly excised tissues. It reveals the morphology of the tissue at the subcellular level within minutes without any calibration or parameters set by users. There is no need for fixation, embedding, nor mounting of the excised tissues on slides. A digital image of the specimen surface is displayed in artificial purple coloring on a touch screen, which allows the surgeon to zoom in and move around the image. The HS version used is CE, marketed since October 2018, and available to accommodate large biological specimens such as lumpectomies (Figure 1).

The surgical team dedicated to breast-conserving surgery at our certified breast center consists of a surgeon performing the procedure and a senior surgeon overseeing the operation and conducting intraoperative assessments. We integrated the HS assessment into our routine practice by having the senior surgeon perform it while the 2nd surgeon prepared the patient for closure. This workflow ensures that the use of the HS does not extend the overall surgical time.

### 2.3. Training of the Surgeons on HS Images

One surgeon experienced in breast surgery of the Gynecology Department of HFR Fribourg and part of the certified Breast Center team (Dr. Guani, a member of the Core Team of the Fribourg Breast Cancer Center) participated in training prior to using the HS. The surgeon completed an online training program divided in two parts. The first part focused on reading sheets of Regions of Interests (ROIs) in HS images. This part is composed of 13 sessions on the features of normal and cancerous breast tissues, including training and self-assessment after training, all performed on-line autonomously by the surgeon. The surgeon can follow the second part of the program when self-assessment performances reach 90% accuracy. This second part contains full HS images of breast cancer cases. This part is divided in three sessions of reviewing annotated and blank images in autonomy complemented with a review meeting with an external expert in HS images from SamanTree Medical. The surgeon completes the program when the performance reaches 80% accuracy for the determination of the margin status in the second phase (Table 1).

### 2.4. Intra-Operative Margin Analysis

The surgical specimen is excised from the patient following the standard-of-care practice of our certified Breast Cancer Center and includes very limited usage of an electric knife. Following excision, the specimen is assessed with the HS. First, the specimen is gently swabbed with a paper towel to remove excess blood, immersed in a fluorescent contrast agent (Histolog Dip, SamanTree Medical, Switzerland) for 10 s, rinsed briefly with saline, and gently swabbed again with a paper towel. Then, the specimen is placed over the imaging window of the HS preliminarily covered with a disposable protective dish. After 50 s, a digital image of the excision surface is displayed, allowing the trained surgeon to review its content for cancerous lesions (Figure 2). The excision is not further processed for HS imaging and remains intact for a later final pathological assessment. Lumpectomy specimens that benefit from an intra-operative mammographic check-up are sent to radiology as usual to verify the presence of the biopsy clip and then sent to pathology to follow standard-of-care procedures for postoperative assessments. Operative specimens are sent to an external pathology lab and the final pathology report is received several days later.

### 2.5. Statistical Analysis

The following covariates were collected and analyzed: patient demographics (age), tumor-related characteristics (tumor type—invasive, in situ, mixed; histological subtype—ductal, lobular, other rare subtypes; tumor size), treatment-related factors (neoadjuvant therapy), and specimen characteristics (length, width, and thickness). The margin status on HS images and final pathology served as the basis for calculating sensitivity, specificity, accuracy, positive predictive value (PPV), and negative predictive value (NPV).

We evaluated the sensitivity and specificity of the HS assessment in relation to the gold standard, the final histopathological analysis. We computed the positive predictive value (PPV), negative predictive value (NPV), and overall accuracy. The 95% confidence intervals (CIs) for all values were computed using Wilson’s formula. We calculated means and percentages (with range) for patient and tumor characteristics and lumpectomy sizes. All statistical analyses were conducted using StatPlus:mac, a statistical analysis software developed by AnalystSoft Inc., version v8 for macOS.

## 3. Results

### 3.1. Training on HS Images

Training is composed of two phases. The purpose of the first phase is to understand main morphological features of normal and cancerous breast tissues in HS images. This phase is composed of eleven sessions presenting HS breast image ROIs provided with normal and cancerous annotations for learning or provided free of annotations for performance assessments. All the sessions of this phase were completed in 1 h 14 min (mean per session 6 min 45 s, min 3 min 20 s, max 14 min 36 s) within three consecutive days with an average of 3.66 sessions per day (min 2, max 6). Accuracy scores achieved in assessment sessions were either 90% or 100%, allowing the surgeon to continue with the second phase of the training. The purpose of this next phase is to understand how to screen full HS images and efficiently recognize the main morphological features of normal and cancerous areas in lumpectomy margins. This phase is composed of three sessions presenting full HS images provided with normal and cancerous annotations for learning or provided free of annotations for performance assessments. In addition, one meeting with an external HS image expert took place to review user annotations and provide additional guidance. The two sessions prior to the meeting were performed for 2 h 17 m in one day with a 55% accuracy score. The consecutive 2 h meeting with a HS expert clarified the image assessment process, including recommended magnification levels for screening and criteria for content evaluation, as well as image contrast adjustments. Following this meeting, a second blind assessment of blank images was performed. The corresponding accuracy score was improved to the recommended value of 80% for using the HS for lumpectomy margin assessments (Table 2).

### 3.2. Patient and Tumor Characteristics

A total of 68 patients were retrospectively analyzed. They represented a population of consecutive patients with breast cancer treated with breast-conserving surgery by the surgeon trained on the HS. The mean age of the population was 60.4 years (29–91 years) and mean size of the lesions defined postoperatively was 0.47 cm (standard deviation 0.50 cm). The mean length, width, and thickness of the lumpectomy specimens were 6.3, 6.2, and 3.0 cm, respectively. A total of 41.2% of the patients were confirmed to have pure invasive tumors, while in situ carcinoma was found in 58.8% of the patients as either pure (11.7%) or mixed with invasive tumors (47.1%). The ductal carcinoma subtype was observed in 64.7% of the patients, while lobular and rarer subtypes (e.g., mucinous, micropapillary, etc.) were found in 19.1% and 16.2%, respectively. A small proportion of the population (13.2%) underwent neoadjuvant treatment prior to breast-conserving surgery (Table 3). The types of BCS were either primary (66/68 patients, 97.1%) or re-excision (2/68, 2.9%). No adverse events were reported intraoperatively or by the final pathology assessments that included H&E staining and immunohistochemistry techniques.

### 3.3. Intraoperative Assessment and Re-Excision Rate

Intraoperative assessments in our certified Breast Center include HS imaging for all patients. For a non-palpable lesion, a harpoon wire is placed preoperatively, and these specimens also undergo mammography. The HS is used to evaluate cancerous status of the specimen margins, while the purpose of the mammogram is to confirm the presence of the biopsy clip in the excised specimen. The mammogram is also interpreted by the surgeon. Intraoperative mammograms of the specimens of fourteen patients in our cohort were reviewed by the same surgeon.

Due to our implementation setting with one operating surgeon and one supervising surgeon during breast-conserving surgeries, the intraoperative assessment of lumpectomy margins with HS does not increase surgical times. Intraoperative recuts were not analyzed with the HS.

The performances of the surgeon for detecting breast cancer in HS images of lumpectomy margins were quantified by comparing the margin status determined intraoperatively with the final pathology assessment. This comparison defines if the HS assessments were true positive, true negative, false positive, or false negative for the margin status. The corresponding sensitivity, specificity, accuracy, positive predictive value, and negative predictive values of breast cancer detected by surgeons using the HS were 100.0%, 96.3%, 96.9%, 85.7%, and 100.0%, respectively (Table 4).

None of the re-excisions in our series were performed because of a missing clip or invisible lesion on mammograms of the specimens; all were based on the HS analysis.

Twelve patients (17.6%) had cancer-positive margins on the primary excision. The types of the lesions localized in these positive margins were pure NST invasive carcinoma (6/12), pure ductal carcinoma in situ (3/12), NST mixed with DCIS (2/12), and micropapillary invasive carcinoma (1/12) (Figure 3). All the positive margins were accurately detected intraoperatively by the surgeon using the HS, triggering the excision of additional tissue during surgery. The additional excised tissues were sufficient to achieve a cancer-negative status in all patients, resulting in no re-excisions in our patient population (Table 5). The surgeon conducted a false positive assessment of HS images for two patients. Retrospective review of the related margins showed roundish structures of fibrous connective tissue mistaken for DCIS. All patients that received neoadjuvant treatment were correctly evaluated with the HS, with one true positive and eight true negatives. At the time of our analysis, the impact of the HS on long-term oncological outcomes showed no recurrence (range from 6 months to 2.5 years).

## 4. Discussion

We report, for the first time in the literature, the implementation of the Histolog Scanner in the routine practice for real-time margin assessments during BCS. An experienced surgeon from our Breast Center benefited from the training, which lasted approximately 6 h, spread over six days, before implementing the HS in daily practice. The results of our retrospective analysis show that use of the HS by the surgeon provides a highly accurate intraoperative margin assessment during BCS, with a sensitivity of 100.0%, resulting in the absence of re-excision and oncological recurrence.

One of the major challenges in breast-conserving surgery (BCS) is to achieve negative surgical margins while preserving as much healthy breast tissue as possible. Conventional intraoperative margin assessment techniques, including frozen section analyses and specimen mammography, have notable limitations in terms of both time requirements and diagnostic accuracy [22,23,24].

Intraoperative analysis of frozen sections has been shown to significantly reduce reoperation rates [25], but it is often time- and resource-consuming. For non-palpable lesions, a guidewire or wireless marker is typically placed. According to a meta-analysis [26], intraoperative specimen mammography demonstrates a sensitivity of 0.55 (95% CI, 0.47–0.63) and a specificity of 0.85 (95% CI, 0.78–0.90). The use of two-view intraoperative specimen mammography can reduce the reoperation rate by up to 50%, as it improves the detection of inadequate margins compared with single-view imaging [27]. 

Our center does not have an on-site pathologist. As a result, we do not routinely perform frozen section analyses; operative specimens are sent to an external pathology laboratory, and the final pathology report is only available several days later. The lack of on-site pathologists is a common issue in many hospitals. It prolongs the operative time whenever a frozen section is required, as the specimen must be transported to the laboratory for analysis. In our case, the laboratory is located 45 min away by car.

We routinely perform intraoperative specimen mammography; however, wire localization, despite being the current gold standard [28], is associated with several limitations.

This technique is usually limited to one or two planar projections, providing only two-dimensional images of the excised tissue. Although this method is useful for confirming the presence of clips or calcifications, it lacks true three-dimensional visualization, and the margin assessment remains imprecise due to tissue superposition and positioning. Compared with techniques such as confocal (HS) imaging, its ability to accurately evaluate all margins is therefore limited.

We propose a model in which the surgeon performs intraoperative margin assessment of the lumpectomy specimen directly using the confocal microscope. An alternative approach to the HS margin assessment could involve an analysis performed by pathologists rather than surgeons. This strategy may be feasible in centers with on-site pathology support. For better context, Table 6 summarizes the main published studies on the HS, including our results, highlighting differences in operator type, diagnostic performance, and impact on re-excision rates.

Our study confirms that confocal laser microscopy (HS) offers a rapid and reliable alternative, allowing a real-time assessment without prolonging the surgical time in our setting. Integration of the HS into routine clinical practice was achieved without disrupting the surgical workflow. No adverse events during and after surgery and no impact on the final pathology assessment have been observed, as previously reported [15,16,17,18,19,20,21,29]. All the HS images produced were assessable by surgeons in real time thanks to the very limited usage of the electric knife and an efficient surgical workflow, allowing HS imaging of the specimen a few minutes after excision, as previously reported [16,17]. Limiting the use of the electric knife is needed in the HS approach. This excision tool will create a burned and hard surface of the lumpectomy that will limit its microscopic assessment [16,17]. Therefore, excising lumpectomies with cold scalpel/scissors and only using cauterization to coagulate the vessels is recommended in order to achieve the best benefit for the patient using this approach. Similarly, cautery is also known to negatively impact assessments of standard histology in some cases. By assigning the HS evaluation to the supervising surgeon, while the operating surgeon completed potential lymph node excision and prepared the patient for closure, we ensured optimal efficiency. This setup minimized the time and allowed for seamless intraoperative decision-making. In a setting with only one surgeon, the time to evaluate all lumpectomy margins with the HS was reported to be around 13 min, which is considered an acceptable amount of time considering the benefit to the patient with this approach [21]. In addition, we agree with Lux et al. that the imaging step of the HS approach can be delegated to some trained medical staff, reducing the amount of additional time required. Prior to the release of the training program described in the present article, 27.27% to 37.5% sensitivities were reported in the literature for the detection of breast cancer in lumpectomy margins by surgeons using the HS [16,17,18]. When using the training program, 81% and 100% sensitivities were achieved by surgeons in the SHIELD study and in our study, respectively [21]. This highlights the need for efficient training material for surgeons that are not experienced in reading microscopic images. The learning curve for HS interpretation was short and acceptable for medical practice, as surgeons were quickly comfortable with the HS imaging patterns. In our opinion, the new training program and HS breast tissue image atlas will also play key roles in standardizing interpretation skills, further supporting the use of the HS for widespread clinical adoption. So far, patients with neoadjuvant treatment and re-excisions were excluded from the cohorts of previous studies of the HS in BCS [15,16,17,21]. Such treatment(s) can make histological readings more complex. Interestingly, the assessment of these patients with the HS has not created false assessments in our patient population, since all of them were accurately assessed with the HS.

### Limitations

Limitations of the present study include the limited sample size, limiting statistical power, especially for rare subtypes such as micropapillary carcinoma. Another limitation of our study is its retrospective design without a control arm. In addition, patients come from a monocentric setting, and the HS approach was performed by one breast surgeon.

Assessment of oncological recurrence was not an objective of this study. A longer follow-up period will be required to evaluate disease-free survival and long-term oncological outcomes.

To validate the external reproducibility and safety of these findings across different surgical teams and patient populations, these promising results need to be confirmed through further prospective studies including a control arm and conducted in a multicenter setting. Strengthening external validity and assessing the generalizability of the findings can facilitate a more appropriate translation of research outcomes into clinical practice [30]. Our center is currently preparing a prospective clinical study to confirm the present results.

## 5. Conclusions

Our study highlights the high diagnostic accuracy and clinical feasibility of using the Histolog Scanner in intraoperative margin assessments during BCS. These findings suggest that it is a valuable tool for optimizing surgical outcomes and improving patient care. By significantly limiting the number re-excisions, the Histolog Scanner represents a valuable addition to standard surgical practice, improving both oncological safety and patient outcomes. 

## Figures and Tables

**Figure 1 cancers-17-02852-f001:**
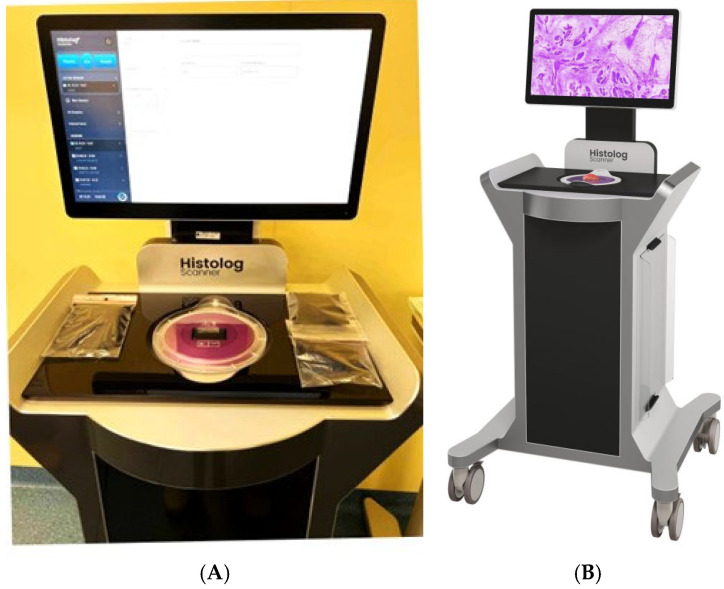
Histolog^®^ Scanner (**A**) Picture in the operating room of HFR Fribourg. (**B**) Illustration in the in-struction manual provided by SamanTree Medical.

**Figure 2 cancers-17-02852-f002:**
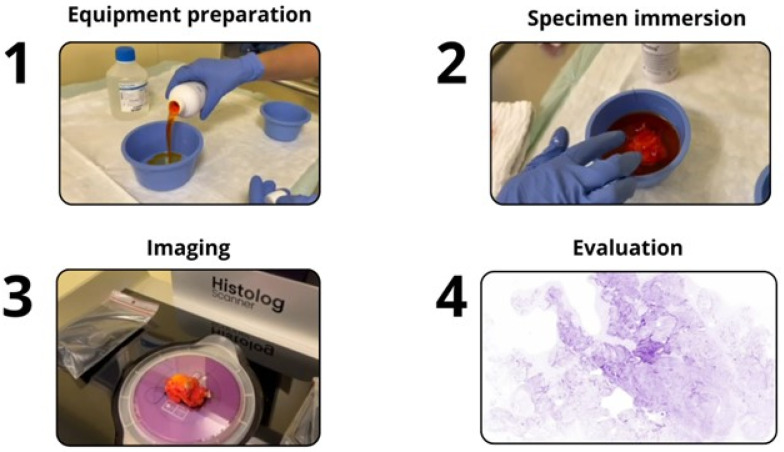
Histolog^®^ Scanner workflow during surgery.

**Figure 3 cancers-17-02852-f003:**
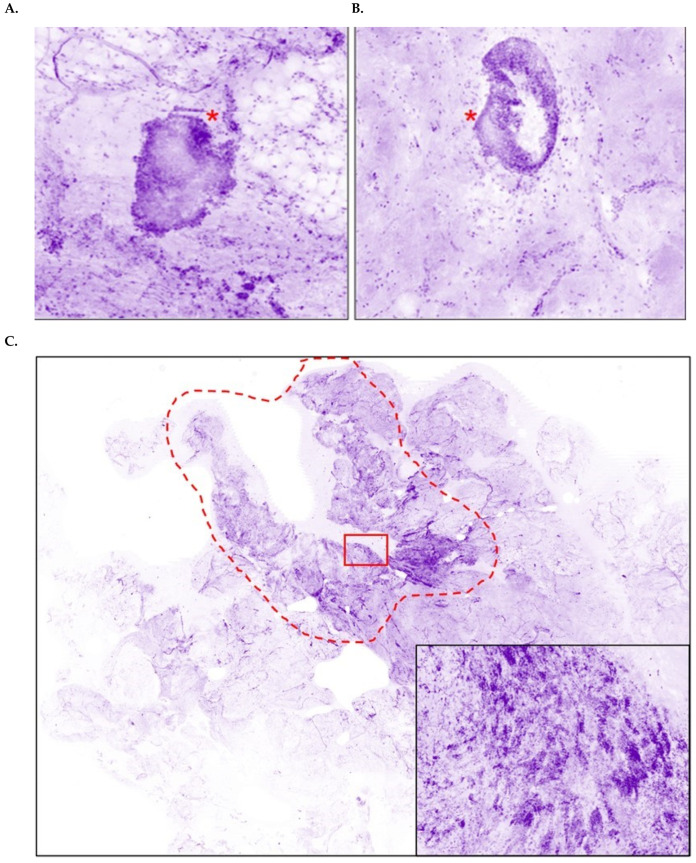
Histolog images of positive breast cancer margins. (**A**, **B**) DCIS lesions (*) seen as solid assemblies of cancer cells presenting an epithelial pattern and surrounded by normal parenchyma (high magnification). (**C**) IDC NST positive margin (low magnification). Insert: nests of cancer cells seen at higher magnification. (**D**) ILC positive margin (low magnification). Insert: trabecular pattern of the tumor seen at higher magnification.

**Table 1 cancers-17-02852-t001:** HS breast image E-learning program.

Phase	Session	Type	Content
P1	S1	ROIs	Normal breast structures 1 (n = 15)
	S2	ROIs	Normal breast structure: Self-assessment (n = 10)
	S3	ROIs	Invasive ductal carcinoma: Training (n = 15)
	S4	ROIs	Invasive ductal carcinoma: Self-assessment (n = 10)
	S5	ROIs	DCIS: Training (n = 15)
	S6	ROIs	DCIS: Self-assessment (n = 10)
	S7	ROIs	Invasive lobular carcinoma: Training (n = 15)
	S8	ROIs	Invasive lobular carcinoma: Self-assessment (n = 10)
	S9	ROIs	Summary: Self-assessment 1 (n = 15)
	S10	ROIs	Summary: Self-assessment 2 (n = 15)
	S11	ROIs	Summary: Self-assessment 3 (n = 15)
P2	S12	HS images	Annotated margins with IDC, DCIS, and ILC (n = 5)
	S13	HS images	Blind assessment of blank margins 1 (n = 9)
	S14	HS images	Review meeting with an external HS image expert
	S15	HS images	Blind assessment of blank margins 2 (n = 10)

**Table 2 cancers-17-02852-t002:** Performance and workload of the surgeon training on HS images.

Session	Content	Workload	Day	Score
P1-S1 (ROI)	Normal breast structures	7 min	Day 1	-
P1-S2 (ROI)	Normal breast structure: Self-assessment	3 min 45 s	Day 1	90%
P1-S3 (ROI)	IDC: Training	7 min 31 s	Day 2	-
P1-S4 (ROI)	IDC: Self-assessment	4 min 12 s	Day 2	90%
P1-S5 (ROI)	DCIS: Training	9 min 27 s	Day 2	-
P1-S6 (ROI)	DCIS: Self-assessment	3 min 52 s	Day 3	100%
P1-S7 (ROI)	ILC: Training	6 min 38 s	Day 3	-
P1-S8 (ROI)	ILC: Self-assessment	3 min 20 s	Day 3	100%
P1-S9 (ROI)	Summary: Self-assessment 1	14 min 36 s	Day 3	100%
P1-S10 (ROI)	Summary: Self-assessment 2	5 min 47 s	Day 3	100%
P1-S11 (ROI)	Summary: Self-assessment 3	4 min 09 s	Day 3	100%
*Total ROIs*	*-*	*1 h 14 min*	*3 days*	*-*
P2-S12 (HS images)	Annotated margins with IDC, DCIS, and ILC	31 min 37 s	Day 1	-
P2-S13 (HS images)	Blind assessment of blank margins 1	1 h 46 min	Day 1	55%
P2-S14 (HS images)	Review meeting with an HS image expert	2 h	Day 2	-
P2-S15 (HS images)	Blind assessment of blank margins 2	49 min 32 s	Day 3	80%
*Total* *HS images*	*-*	*5 h 08 min*	*2 days*	*-*

**Table 3 cancers-17-02852-t003:** Population characteristics.

Characteristic	Value
Age (mean, range) [years]	60.4 (29–91)
Breast cancer type Invasive carcinoma In situ carcinoma Invasive with in situ carcinoma	41.2% (28/68) 11.7% (8/68) 47.1% (32/68)
Carcinoma subtypes Ductal Lobular Other	64.7% (44/68) 19.1% (13/68) 16.2% (11/68)
Tumor size (mean ± standard deviation) [cm]	0.47 ± 0,50
Neoadjuvant treatment	13.2% (9/68)
Type of breast-conserving surgery Primary Re-excision	97.1% (66/68) 2.9% (2/68)
Specimen sizes (mean, range) [cm] Length Width Thickness	6.3 (3–10) 6.2 (2.4–10) 3.0 (1.5–7.3)

**Table 4 cancers-17-02852-t004:** Intraoperative cancer detection performance of surgeons during lumpectomy. Margins status were determined by the surgeon based on HS images content. The 95% confidence interval is shown in brackets.

Parameter	Value
Sensitivity	100.0% (75.8–100.0%)
Specificity	96.3% (87.5–98.9%)
Accuracy	96.9% (89.6–99.2%)
PPV	85.7% (60.1–95.9%)
NPV	100.0% (93.1–100.0%)

**Table 5 cancers-17-02852-t005:** Re-excision rate, recurrence, and cancer type for positive margins after primary excision.

Parameter	Value
**Re-excision rate** (95% CI)	0% (0–12.7%)
**Breast cancer type based on the margins** NST NST and DCIS DCIS Micropapillary	6/12 2/12 3/12 1/12
**Recurrence** (95% CI) (6 months–2.5 years of follow-up)	0% (0–7.7%)

**Table 6 cancers-17-02852-t006:** Comparative results of studies of the Histolog Scanner (HS).

Study	Year	Setting/Operator	N (Patients/Images)	Sensitivity (%)	Specificity (%)	Accuracy (%)	PPV (%)	NPV (%)	Re-Excision/Re-Operation	Notes
Cattacin et al. (this submission)	2025	Single-center, retrospective; surgeon-operated HS	68 patients	100.0	96.3	96.9	85.7	100.0	0% re-excisions in the cohort	HS integrated in the routine workflow
SHIELD (Lux et al.) [21]	2025	Prospective, single-center; surgeon-operated HS	50 patients	80.95	99.51	97.8	94.44	98.09	10% vs. 30% historical (*p* = 0.016)	Substantial reduction in re-operations vs. the historical control
Colard-Thomas et al. [18]	2025	Single-center, retrospective; pathologist-operated HS	20 patients	100.0	92.3	95.2	88.9	100.0	0% re-excisions in the cohort	HS integrated in the routine workflow
HIBISCUSS (Conversano et al.)–**Surgeons** (Round 7) [15]	2023	Training/validation of an image library; surgeons	—/300 images	97.0	99.0	98.0	-	-	—	Image-level performance (not an intraoperative margin study)
HIBISCUSS (Conversano et al.)–**Pathologists** (overall) [15]	2023	Training/validation of an image library; pathologists	—/300 images	98.0	99.6	99.6	-	-	—	Image-level performance (not an intraoperative margin study)

## Data Availability

The data presented in this study are available upon request from the corresponding author. The data are not publicly available due to privacy and ethical restrictions.

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
