# Peer review of "Confocal Microscopy for Intraoperative Margin Assessment of Lumpectomies by Surgeons in Breast Cancer: Training, Implementation in Routine Practice, and Two-Year Retrospective Analysis"

_cancers, 2025, doi:10.3390/cancers17172852_

Round 1
Reviewer 1 Report
Comments and Suggestions for Authors
This study is the first to report real-world, two-year clinical experience using the Histolog® Scanner (HS) for intraoperative margin assessment in breast-conserving surgery (BCS), and it does offer valuable insights. However, several concerns remain. The study included only 68 patients, limiting statistical power, especially for rare subtypes such as micropapillary carcinoma. The results may be influenced by the surgeon’s experience and device-handling habits, raising questions about external reproducibility. Additionally, no head-to-head comparison was made with traditional methods such as frozen section analysis or intraoperative mammography, making it difficult to quantify the magnitude of HS’s advantages. Finally, the median follow-up of only 6 months to 2.5 years is insufficient to assess long-term recurrence rates.
Author Response
We sincerely thank the reviewer for these valuable and constructive comments, which helped us clarify and improve the manuscript.
We respond point by point below:
Reviewer’s comment 1:
“The study included only 68 patients, limiting statistical power, especially for rare subtypes such as micropapillary carcinoma.”
Response:
We acknowledge that the sample size is relatively small, indeed, we have already planned a prospective study with a larger cohort to confirm and strengthen these preliminary results. In fact this study is a retrospective analysis with the primary aim of assessing the feasibility of introducing the Histolog® Scanner (HS) for intraoperative use by surgeons during routine lumpectomies. Our focus was on evaluating the quality of the training process and the surgeons’ performance after training. This point has now been clarified in the Materials and Methods section (2.1- Study design ) and is further discussed in the Discussion section (4.1 - Limitations).
Reviewer’s comment 2:
“The results may be influenced by the surgeon’s experience and device-handling habits, raising questions about external reproducibility.”
Response:
We agree with this comment. The results may indeed be influenced by the surgeon’s learning curve and handling habits. However, external reproducibility was not the objective of this retrospective analysis. Our primary aim was to evaluate the feasibility and performance of the HS after a structured training program within our single-center setting. Notably, two recent studies (ref. 11 and 14, 2025) have reported results similar to ours, which supports the consistency of our findings despite the limited scope. Nevertheless, we fully acknowledge that reproducibility should be further assessed in future multicenter prospective studies including surgeons from different institutions, as this would allow evaluation of performance across varying levels of experience and practice.
Reviewer’s comment 3:
“No head-to-head comparison was made with traditional methods such as frozen section analysis or intraoperative mammography, making it difficult to quantify the magnitude of HS’s advantages.”
Response:
We initiated this study because our institution does not have on-site pathology. As a result, we do not routinely perform frozen section analysis; operative specimens are sent to an external pathology lab and the final pathology report is received only several days later. The lack of on-site pathologists is a common issue in many hospitals.
We do, however, regularly perform intraoperative specimen mammography to verify the presence of the biopsy clip, but the mammography is also interpreted by the surgeon. As stated in line 230 of the manuscript: “In the case of a non-palpable lesion, a harpoon wire localization is placed preoperatively, and these specimens will also undergo a specimen mammography.” Fourteen patients in our cohort had intraoperative specimen mammography reviewed by the same surgeon. None of the re-excisions in our series were performed because of a missing clip or invisible lesion on specimen mammography; all were based on HS analysis.
It is also important to note that intraoperative specimen mammography consists of a single frontal projection and does not provide three-dimensional visualization of the specimen, making it difficult to accurately assess surgical margins compared to HS imaging.
We have clarified this point in the Materials and Methods section (2.4 - Intra-operative margins analysis) and the Results section (3.4 - . Intraoperative assessment and re-excision rate).
Reviewer’s comment 4:
“Finally, the median follow-up of only 6 months to 2.5 years is insufficient to assess long-term recurrence rates.”
Response:
We agree with this comment. We are aware that the follow-up period in our study is too short to assess long-term recurrence rates or disease-free survival. As recurrence assessment was not an objective of this retrospective analysis, a longer follow-up will be needed to address this point in future studies.
We have explicitly added this limitation and corresponding explanation in the Discussion section (4.1 – Limitations).
-----------------------------------------------------------------------------------------------------------------------These changes and clarifications have been incorporated into the revised manuscript, specifically in sections 2.1, 2.4, 3.4, and 4, and we believe they better explain the scope and limitations of our study and will guide future research directions. In addition to the specific points addressed above, we have revised and enriched the Introduction and Research Design, as well as refined the Conclusions, in order to optimally address the reviewer’s comments.
Reviewer 2 Report
Comments and Suggestions for Authors
This study addresses an interesting topic; however, the manuscript would benefit from several improvements.
The authors should clearly state their hypothesis in the introduction.
Additionally, the primary endpoint should be explicitly defined, and the covariates analyzed should be described in detail.
The organization of the paper could be improved, with the information presented in a more systematic and scientifically rigorous manner.
The limitations need to be better described together with suggestions for future studies on this topic.
Author Response
We sincerely thank the Reviewer for these constructive and valuable comments, which have helped us improve the clarity and scientific rigor of our manuscript.
We respond point by point below:
Reviewer 2 – Comment 1:
“The authors should clearly state their hypothesis in the introduction.”
Response:
We thank the reviewer for this remark. We have added a clear statement of our study hypothesis in the Introduction section of the revised manuscript.
Reviewer 2 – Comment 2:
“Additionally, the primary endpoint should be explicitly defined, and the covariates analyzed should be described in detail.”
Response:
We agree with this important comment. We have explicitly specified the primary and secondary outcomes in the Materials and Methods section (2.1), along with a detailed description of the analyzed covariates.
Reviewer 2 – Comment 3:
“The organization of the paper could be improved, with the information presented in a more systematic and scientifically rigorous manner.”
Response:
We appreciate this helpful suggestion. We have reorganized the structure of the manuscript and reformulated key parts of the Introduction, Materials and Methods, and Results sections to ensure a clearer and more systematic presentation, consistent with scientific standards.
Reviewer 2 – Comment 4:
“The limitations need to be better described together with suggestions for future studies on this topic.”
Response:
We agree with the reviewer’s point. We have added a dedicated subsection (4.1) in the Discussion addressing the study’s limitations and providing suggestions for future research.
------------------------------------------------------------------------------------------------------------------------------------
Once again, we thank Reviewer 2 for these thoughtful and constructive comments. They have greatly contributed to improving the clarity, organization, and scientific quality of our manuscript.
Reviewer 3 Report
Comments and Suggestions for Authors
Many thanks to the authors for giving me the opportunity to review this manuscript. It is very well written and clear. The subject is interesting.
I have some comments and questions:
Chapter introduction
Line 73 – 74 you written : The high rate of re-excisions leads to additional costs, increased patient anxiety, and a higher risk of postoperative complications.
- You can add that often the cosmetic result is degraded after re-excision.
- Do you know your re-excision rate prior to your study?
Chapter results
- Line 253 for the additional tissue recuts, did you use the histolog scanner again to check for the absence of tumor cells on the new margin?
Chapter discussion
- Two surgeons are present for this surgery, one for the surgery and the other for using the histolog scanner. In the Hibiscuss project, pathologists were trained in the use of the histolog scanner. Is this a viable alternative?
Breast carcinoma detection in ex vivo fresh human breast surgical specimens using a fast slide-free confocal microscopy scanner: HIBISCUSS project
Angelica Conversano et al., BJS Open 2023 May 5;7(3):zrad046. doi: 10.1093/bjsopen/zrad046.
- Can you make a table summarizing the results of the main studies and your results?
Author Response
We sincerely thank the Reviewer for the constructive and insightful comments, which helped us further improve the manuscript. Please find our point-by-point responses below:
Reviewer 3 – Comment 1 (Chapter Introduction):
“Line 73–74: You wrote: ‘The high rate of re-excisions leads to additional costs, increased patient anxiety, and a higher risk of postoperative complications.’ You can add that often the cosmetic result is degraded after re-excision.
Do you know your re-excision rate prior to your study?”
Response:
We agree with this valuable suggestion. We have added in the Introduction (section 1) that re-excisions often negatively affect cosmetic outcomes.
Furthermore, we have specified our institutional re-excision rate prior to the study: it was 5%, in line with the recommendations of the Swiss National League Against Cancer among the accreditation criteria for breast centers in Switzerland. This information has been added to provide better context in the Introduction.
Reviewer 3 – Comment 2 (Chapter Results):
“Line 253: For the additional tissue recuts, did you use the Histolog® Scanner again to check for the absence of tumor cells on the new margin?”
Response:
We thank the reviewer for this pertinent question. We did not systematically re-analyze the additional recuts with the HS in this retrospective study; however, this evaluation is planned for the prospective study we intend to launch with a larger sample size. This point has been clarified in the Materials and Methods section (2.4).
Reviewer 3 – Comment 3 (Chapter Discussion):
“Two surgeons are present for this surgery, one for the surgery and the other for using the Histolog® Scanner. In the Hibiscuss project, pathologists were trained in the use of the Histolog® Scanner. Is this a viable alternative?
Breast carcinoma detection in ex vivo fresh human breast surgical specimens using a fast slide-free confocal microscopy scanner: HIBISCUSS project (Conversano et al., BJS Open 2023;7(3):zrad046).
Can you make a table summarizing the results of the main studies and your results?”
Response:
We initiated this study because our institution does not have on-site pathology. As a result, operative specimens are sent to an external pathology lab and the final pathology report is received only several days later. The lack of on-site pathologists is a common issue in many hospitals.
We have discussed the possibility of alternative approaches, such as the Hibiscuss project model, in the Discussion. Additionally, as suggested, we have added a table summarizing the main published studies (including Hibiscuss) and our results for better comparison.
-----------------------------------------------------------------------------------------------------------------------
Once again, we thank the Reviewer for these thoughtful comments, which allowed us to improve the completeness and clarity of our manuscript.
Round 2
Reviewer 1 Report
Comments and Suggestions for Authors
I have no other problems
Reviewer 2 Report
Comments and Suggestions for Authors
I think that most of revisions and suggestions have been addressed.